# Compact Exposimeter Device for the Characterization and Recording of Electromagnetic Fields from 78 MHz to 6 GHz with Several Narrow Bands (300 kHz)

**DOI:** 10.3390/s21217395

**Published:** 2021-11-07

**Authors:** Marco Xavier Rivera González, Nazario Félix González, Isabel López, Juan Sebastián Ochoa Zambrano, Andrés Miranda Martínez, Ceferino Maestú Unturbe

**Affiliations:** 1Center for Biomedical Technology (CTB), Universidad Politécnica de Madrid (UPM), 28040 Madrid, Spain; Nazario.felix@ctb.upm.es (N.F.G.); isabel.lopezd@alumnos.upm.es (I.L.); andres.miranda@ctb.upm.es (A.M.M.); ceferino.maestu@ctb.upm.es (C.M.U.); 2School of Computer Systems Engineering (ETSISI), Universidad Politécnica de Madrid (UPM), 28040 Madrid, Spain; js.ochoa@alumnos.upm.es; 3Networking Research Center of Bioengineering, Biomaterials and Nanomedicine (CIBER-BBN), Universidad Politécnica de Madrid (UPM), 28040 Madrid, Spain

**Keywords:** exposimeter, electromagnetic fields, power density, radiofrequency

## Abstract

A novel compact device with spectrum analyzer characteristics has been designed, which allows the measuring of the maximum power received in multiple narrow frequency bands of 300 kHz, recording the entire spectrum from 78 MHz to 6 GHz; the device is capable of measuring the entire communications spectrum and detecting multiple sources of electromagnetic fields using the same communications band. The proposed device permits the evaluation of the cross-talk effect that, in conventional exposimeters, generates a mistake estimation of electromagnetic fields. The device was calibrated in an anechoic chamber for far-fields and was validated against a portable spectrum analyzer in a residential area. A strong correlation between the two devices with a confidence higher than 95% was obtained; indicating that the device could be considered as an important tool for electromagnetic field studies.

## 1. Introduction

With the advancement of technology, the population is increasingly exposed to radiation by non-ionizing electromagnetic fields in the radio frequency range (RF-EMF), which is from 100 kHz to 300 GHz; these frequencies correspond to wireless technologies, such as Wifi, mobile phone, television, radio, etc.

According to global statistics reported by the International Telecommunication Union (ITU), carried out from 2001 to 2018, there is a constant growth in the use of wireless technologies [1], which has generated concern in society about the possible health effects associated with RF-EMF exposures [2]. The regulatory organizations limit the amount of radiated energy following the International Commission on Non-Ionizing Radiation Protection (ICNIRP) guidelines that only consider the so-called thermal effects using the criterion of the maximum rate of radiation and Specific Absorption Rate (SAR) as the only evaluation parameter, which consists of the measurement of the maximum RF-EMF power absorbed by living tissue [3]. Nevertheless, the effects that could be considered non-thermal are not recognized within the scientific evidence by ICNIRP [3,4,5,6,7,8,9]. The recommendations of the ICNIRP establish the maximum levels of radiation at 450 µW/cm^2^ [3,9,10,11]. However, the BioInitiative working group, together with other researchers [8,10,12,13,14,15,16], suggest that adverse health effects are observed at low levels of exposure 0.1 µW/cm^2^. Studies suggest that RF-EMF exposures with powers below the recommendations of the ICNIRP have effects related to changes in brain activity [17], affecting cognitive and motor performance [12,13], infertility problems in the male reproductive system [18,19], DNA damage [20,21], association to different brain tumors and intensity of RF-EMF, and having a greater effect in children and teenagers than in adults [4,6,12,22,23,24,25]. These studies suggest that exposure to RF-EMF is an important factor to consider as a “possible carcinogen” classified in group 2B by the International Agency for Research on Cancer (IARC) [26]. On the other hand, using the same evaluation criteria, they recommend that RF-EMF exposures should belong to group 2A in the IARC as “probable carcinogenic” [8,15].

The lack of knowledge of the levels of RF-EMF radiation to which the population is exposed raises greater concern due to the possible adverse effects on health resulting from continuous exposure to these electromagnetic fields [1,2,8,12,27]. At the moment, the solution for this problem is the use of systems, such as spectrum analyzers and exposimeters, that allow the measurement and characterization of RF-EMF [28]. According to the criteria of the International Telecommunication Union (ITU), the measurements must be carried out continuously, allowing the power intensity to be measured and discriminate between each frequency corresponding to the RF-EMF emission sources [1]; therefore, measurements made with a spectrum analyzer are the most appropriate for the characterization of RF-EMF. However, it is an expensive method and also requires experts for the correct use and interpretation of the equipment’s data; for this reason, the use of exposimeters is considered as the best option [16]. Different exposimeters perform measurements in broadband [29], which does not allow correct discrimination between the various frequencies and, therefore, the proper identification of the RF-EMF emission sources [30]. There is a clear need to provide to the population a small and low-cost exposimeter that allows monitoring the peak-to-peak maximum electromagnetic radiation levels of the entire radioelectric spectrum in several narrow frequency bands to differentiate between multiple sources in the same band of telecommunications.

## 2. Materials and Methods

The exposimeters used for epidemiological studies of telecommunication bands, such as radio, television, mobile telephony, Wifi, are governed by the regulations and standards for electromagnetic fields, such as the ICNIRP guidelines or IEEE standard C95.1 [3,31], developed by the international committee on Electromagnetic safety [32]. The use of exposimeters arises from spot measurements performed with spectrum analyzers at telecommunication base stations in order to corroborate that the transmitted power is according to the standards. [33]

The most common narrowband exposimeters used in studies are: EME Spy-120, EME Spy-121, EME Spy-140, EME Spy-120 (SATIMO, Courtaboeuf, France), ESM 140 (Maschek Electronik, Bad Wörishofen, Germany), and ExpoM-RF (Fields at Work GmbH, Zürich, Switzerland). All these devices have different characteristics in the number of frequency bands, sampling interval, detector type, size, and weight [29,30]. Table 1 lists the main characteristics of these devices.

Spectrum analyzers allow, with great precision, the exposure to electromagnetic fields both in frequency and power to be determined, allowing the identification of each of the sources of electromagnetic field emission, is a great advantage over the aforementioned exposimeters; however, these have some disadvantages, such as expensive equipment, low precision in dynamic environments, and need expert personnel for their correct operation. On the other hand, exposimeters are cheaper, allowing their use without the need of an expert, are faster in dynamic places, and, due to their size and easy use, allow their implementation in various epidemiological studies [16]. The sensitivity of exposimeters could be improved by performing separate measurements in various fractions of the RF spectrum, where it detects sources that contribute significantly to exposures, considering between 12 and 20 radio frequency bands as the most significant in the entire radio spectrum. However, this is not supported by a systematic investigation of the entire radio frequency spectrum, and it is possible that more bands should be measured or that some of them can be ignored [33].

The major limitation of exposimeters is that these do not allow the measurement of the entire spectrum and are not suitable for differentiating between multiple electromagnetic field sources because its resolution bandwidth is determined by the full desired frequency band to be measured; using a single measurement for the entire selected band avoids adequately showing the user the different sources that occupy the same band [34]. The use of cell phones close to the exposimeter could increase the measured value in the band, and a clear increase, or an overestimation, of the measurement, is observed. [35] Another drawback is cross-talk, a signal that is recorded in a nearby band, providing a false measurement; it usually happens in mobile telephony between Uplink and Downlink bands [30]. The comparison features between conventional exposimeters and spectrum analyzers is shown in Table 2.

The main disadvantage of exposimeters is the inability to measure the entire radioelectric spectrum in several narrow frequency bands in such a way that allows properly differentiating between multiple radiations sources. Therefore, an exposimeter system with similar characteristics to a spectrum analyzer is proposed in this paper without losing the advantages of the traditional exposimeters.

### 2.1. Proposed System

The design and manufacture of the device comprise three phases that are explained below:The first phase covers the design and development of hardware for the detection of electromagnetic (EM) signals in the frequency range between 78 MHz and 6 GHz with bandwidths of 300 kHz.The second phase consists of the calibration of the device in a controlled environment.The third phase is the development of an algorithm for data processing and an interface for visualization of the power density in different frequency bands.

#### 2.1.1. Data Acquisition System

The proposed system consists of an array of fractal antennas (FRACTUSANTENNAS, Barcelona, Spain) to cover the frequency range between 78 MHz and 6 GHz. These antennas are suitable for this design because they can cover large bandwidths and have small dimensions compared to other types of commercial antennas. This makes it possible to optimize space on the printed circuit board (PCB). Table 3 shows the models and bandwidth of the fractal antennas used for the proposed exposimeter. The plots of radiation pattern of each antenna are shown in Appendix B.

Figure 1 shows the general block diagram of the system operation, where the fractal antennas from Table 3 are multiplexed by an Analog Devices HMC321ALP4E RF mux (Analog Devices, DE, USA), which has eight inputs and one output that enables 3:8 binary TTL decoding control; the switching time (*T_S_*) is 125 nsec; and the RF mux switch to the next antenna when the device has measured all its bandwidth. The signals from the antenna are amplified by an Analog Devices ADL5542 low noise amplifier (LNA), which has a maximum gain of 20 dB. The Analog Devices ADF4355-3 fractional-N phase-locked Loop (PLL) works as an RF signal generator, covering the range from 54 to 6600 MHz; the total lock time of the PLL to generate a signal (*T_PLL_*) is 45 µs. The signal generated by the PLL and the signal received from the antennas are multiplied in the mixer in order to lower the frequency of the antenna signal to a frequency that matches the 315 MHz center frequency and 300 kHz bandwidth of the SAW bandpass filter; the Analog devices ADL 5802 mixer and the AFS315E-T filter from Abracon LLC were chosen. The filtered signal is measured by an Analog devices AD8309 logarithmic RF detector that provides a voltage output signal proportional to the power of the input signal in dBm; the output settling time of the logarithmic detector (*T_LD_*) is 220 ns. The output voltage of the logarithmic detector is sampled using an Analog-to-Digital Converter (ADC) and recorded in the microcontroller; the ADC rate (*T_ADC_*) is 1 Msamples/s (1 µs for each sample). The time to measure one sample (*T′*) is described as follows:(1)$$T^{\prime }=T_{PLL}+T_{LD}+T_{ADC},$$
(2)$$T^{\prime }=45\;µs+0.22\;µs+1\;µs=46.22\;µs,$$

The proposed system measures the received power in the spectrum from 78 MHz to 6 GHz with 300 kHz of resolution bandwidth, measuring approximately 19,500 narrow frequency bands. Therefore, the total time to measure all the bands is described as follow:(3)$$T_{T}=n_{S}T^{\prime }+n_{A}T_{S},$$
(4)$$T_{T}=19,500\times 46.22\;µs+5*0.125\;µs=901.915\;µs,$$
where *T_T_* = 901.915 µs is the total time to measure all the bands; *n_S_* is the number of samples (19,500); *n_A_* is the number of antennas; finally, *T_S_* is the switching time of the RF mux.

Finally, the measured signals are stored with the GPS geolocation data in a non-volatile SD memory that can store up to 2 Gbytes of data. The system has a USB communication port to communicate with the computer and download the set of records made. In Figure 2, the flow chart for the measurement process is shown.

The PLL works as a tuner of the signals recorded by the antennas; therefore, consider that the PLL generates a signal *S*_1_, and the antenna array receives a signal *S*_2_. These can be described as follows:(5)$$S_{1}=A_{1}\cos\left(2\pi f_{1}t+\theta_{1}\right),$$
(6)$$S_{2}=kA_{2}\left(2\pi f_{2}t+\theta_{2}\right),$$
where *k* is the amplification value due to the low noise amplifier, *A*_1_ and *A*_2_ are the amplitudes corresponding to each of the signals, and likewise, their frequencies are *f*_1_, *f*_2_ and phases are *θ*_1_ and *θ*_2_, respectively. These are multiplied by the mixer, and the following equation is obtained with the signal mixer theory [36].
(7)$$S_{1}S_{2}=\frac{kA_{1}A_{2}}{2}\left[\cos\left(2\pi \left(f_{1}+f_{2}\right)t+\theta_{1}+\theta_{2}\right)+\cos\left(2\pi \left(f_{1}-f_{2}\right)t+\theta_{1}-\theta_{2}\right)\right],$$

The signal generated by the PLL signal synthesizer is 315 MHz different from the received signal; this allows it to match with the center frequency of the bandpass filter; therefore, *f*_1_ − *f*_2_ is equal to 315 MHz, and Equation (7) would be as follows:(8)$$S_{1}S_{2}=\frac{kA_{1}A_{2}}{2}\left[\cos\left(2\pi \left(f_{1}+f_{2}\right)t+\theta_{1}+\theta_{2}\right)+\cos\left(2\pi \left(315\left(10^{6}\right)\right)t+\theta_{1}-\theta_{2}\right)\right],$$

Then, through the bandpass filter, the signals with a frequency different than 315 MHz are filtered. Giving the next equation:(9)$$S_{1}S_{2}=\frac{kA_{1}A_{2}}{2}\left[\cos\left(2\pi \left(315\left(10^{6}\right)\right)t+\theta_{1}-\theta_{2}\right)\right],$$

Then, by the signal theory [37], the power of the *P_S_*_1*S*2_ signal is expressed with the following equation:(10)$$P_{S_{1}S_{2}}=\underset{T\to \infty }{\lim}\frac{1}{T}{\left(\frac{kA_{1}A_{2}}{2}\right)}^{2}\overset{T}{\underset{-T}{\int }}\cos{\left(2\pi \left(315\left(10^{6}\right)\right)t+\theta_{1}-\theta_{2}\right)}^{2}\partial t,$$
(11)$$P_{S_{1}S_{2}}\left(W\right)=\underset{T\to \infty }{\lim}\frac{1}{T}{\left(\frac{kA_{1}A_{2}}{2}\right)}^{2}T={\left(\frac{kA_{1}A_{2}}{2}\right)}^{2},$$

According to Equation (11), the power of the signal depends only on its amplitude; therefore, the logarithmic detector detects the power of the *P_S_*_1*S*2_ signal expressed in dBm. This value is registered and stored in non-volatile memory using communication with the microprocessor, which also has the function of controlling the RF mux, the PLL, and the GPS to obtain the geolocation of each measurement performed.

As shown in Figure 3, the proposed system consists of two PCBs fitting one on top of the other, incorporating all the components mentioned above. The first PCB (Figure 3a) integrates the radio frequency components; the second PCB (Figure 3b) includes the microcontroller, GPS, SD memory, the components for battery charging, and voltage regulators. The antennas share space between the two PCBs and are connected using coaxial cables. The PCBs were designed with enough free space in the PCB in order to avoid electromagnetic interference; this is specified in the datasheets from FRACTUS ANTENNAS. The system uses 2-cell lithium-ion (Li-ion) batteries of 3500 mAh in series, with low noise linear regulators 3.3 and 5.0 V for power supply. The average current consumption is estimated at 450 mA, in continuous operation mode, allowing the system to run for several hours without the need to charge the batteries.

The proposed system has a sampling rate of 20,000 samples/s, taking less than one second to measure the entire frequency spectrum. The memory provides up to one week of non-volatile memory space. The system has a dynamic power measurement range of 90 dB with RF input power from −70 to +20 dBm and a 0.04 dBm resolution. Table 4 shows the specifications of the proposed system.

#### 2.1.2. System Calibration

The calibration of the device was performed in the anechoic chamber of the Universidad Politécnica de Madrid. The measurements were taken at a distance of 2 m between the transmitting antenna and the proposed exposimeter, evaluating only the far-field from 698 MHz to 6 GHz. The PNA network analyzer (Agilent Technologies E8362B, Santa Clara, CA, USA) was used to generate the signals, and 10 dBm of power was configured. Figure 4 shows a block diagram of the exposimeter calibration system.

As shown in the block diagram in Figure 4, a computer controls the PNA to generate the signals to be measured and communicates with the exposimeter to perform the corresponding measurement, to obtain: first, the data of return losses delivered by the PNA and, second, the measurements of power received by the exposimeter. According to the antenna theory, have the following equations [38]:(12)$$RL=-20log_{10}\left|\Gamma \right|,$$
where *RL* is the return loss expressed in dB generated by the PNA, and *Γ* is the reflection coefficient, which indicates the percentage of the reflected signal. Solving Equation (12), the reflection coefficient is obtained:(13)$$\Gamma =\left(10^{\left(-\frac{RL}{20}\right)}\right),$$

To find the percentage of transmitted power, the following equation was used:(14)$$T=\left(1-10^{\left(-\frac{RL}{20}\right)}\right)$$

In Equation (14), T is the percentage of transmitted power and allows the output power of the transmitting antenna to be calculated. In order to obtain the power received by the antennas, the free space attenuation equation was used [39]:(15)$$L_{bf}=32.4+20log_{10}\left(f\right)+20log_{10}\left(d\right),$$
where *L_bf_* is in dB, *f* is the frequency in MHz, and *d* is the distance in meters. Therefore, the power received by the antennas *P_r_* as a function of the transmitted power *P_T_*, is expressed as follows:(16)$$P_{r}=P_{T}\times T-L_{bf},$$

The signals obtained by the PNA are processed using Equation (16) to obtain the received power value and relate it with the measurements obtained by the exposimeter to create the calibration curves.

#### 2.1.3. Software Development: Data Processing and Visualization System

The stored data by the exposimeter is transmitted to the PC using LabVIEW software (National Instruments Company, Austin, TX, USA) and, the data is processed using Matlab software (MathWorks Company, Natick, MA, USA), where each power measurement is corrected through the calibration wave. To obtain the data in power density (W/m^2^), the following equations according to the theory of radiation parameters were used [40].
(17)$$S=\frac{P_{r}}{A},$$
(18)$$A=\frac{\lambda^{2}}{4\pi }G,$$
(19)Pr(W)=10(Pr(dBm)10)1000,
where *S* is the power density, *P_r_* is the received power in dBm, *λ* is the wavelength expressed in meters, and *G* is the maximum orientation of the antenna that is considered to be 1.5 so that the effective area *A* is a function of the frequency. Therefore, Equation (17) can be written as:(20)S=10(Pr(dBm)10)λ24πG=(4πf2c2)10(Pr(dBm)10),

Equation (20), for power density, is expressed in mW/m^2^ and is a function of the frequency, which is applied to each measure in order to obtain the final result.

### 2.2. Case Study for Validation of the Proposed Design

A case study was carried out to validate the proposed exposimeter against a portable spectrum analyzer. The regulations established by ICNIRP [3] were not followed, the measurements were performed in 10-min periods, and the maximum received power levels were considered (not the average); therefore, the data analysis considers only the peak values or the maximum power recorded at each measurement point. The measurements were performed to validate only the far-field, not on the body, and all measurements were performed outdoors. The validation study was deployed in a residential area in Madrid, Spain; seven different measurement points have been proposed in this urban area. According to the Ministry of Industry, Energy, and Tourism data [41], Figure 5 shows the location of the telephone stations (red) that are near the measurement points (blue). There are nine telephone base stations in this area that operate on the frequencies indicated in Table 5.

The FSH8 portable spectrum analyzer with a TSEMF-B2 omnidirectional antenna was used (Munich, Germany, ROHDE&SCHWARDZ), which allows measurements in bandwidths from 700 MHz to 6 GHz, registering maximum power levels in 10-min intervals. Figure 6 shows the spectrum analyzer and TSEMF-B2 omnidirectional antenna.

The professional equipment (spectrum analyzer) and the proposed exposimeter perform the measurements at each point at the same time. The measures of both devices were carried out at 1.5 m above the ground using a tripod, and personal equipment (mobile phone) was turned off in order to avoid stimulating the signals from the base stations. The proposed exposimeter measured from 78 MHz to 6 GHz; the data from 700 MHz to 6 GHz were analyzed.

Student’s *t*-test was used to evaluate independent variables to verify if there are significant differences between the measurements of the two devices, where the number of samples corresponds to the maximum peak power recorded at each frequency band of the measurement points from 1 to 7.

Then, Pearson’s correlation factor is calculated to find the relationship between the FSH8 spectrum analyzer and the proposed exposimeter, evaluating the linear dependence measure between two variables from the mean and standard deviation of each variable [42], as follows in Equation (21).
(21)$$\rho \left(A,B\right)=\frac{1}{N-1}\overset{N}{\underset{i=1}{\sum }}\left(\frac{A_{i}-{µ}_{A}}{\sigma_{A}}\right)\left(\frac{B_{i}-{µ}_{B}}{\sigma_{B}}\right)$$
where *A* and *B* correspond to the variables to be measured, µ*_A_* and *σ_A_* correspond to the mean and standard deviation of *A*, µ*_B_* and *σ_B_* correspond to the mean and standard deviation of *B*, and finally, *N* corresponds to the number of observations or samples.

## 3. Results

The measurements at each point in Figure 5 were performed with the proposed exposimeter and FSH8 spectrum analyzer at the same time. The plots obtained for each measurement point are shown in Appendix A. Table 6 shows the values of the highest power peaks for each working frequency of the telephone base stations (Table 5) and the frequencies corresponding to 2G and 5G Wifi, where SA and E correspond to the FSH8 Spectrum Analyzer and the proposed Exposimeter, respectively. Finally, the last row indicates the correlation factor between the two pieces of equipment for each measurement point.

Table 7, shows the frequencies corresponding to the maximum peaks of the spectrum analyzer and the exposimeter from Table 6.

The data obtained by the spectrum analyzer are compared with the data registered by the proposed exposimeter using Student’s *t*-test. The Kolmogorov Smirnov test has been used to verify the normality of the measurements. The number of samples (N = 49) corresponds to the maximum power recorded at each frequency band of the measurement points from 1 to 7 in Table 6. The resulting data is transformed into logarithmic units, obtaining the results presented in Table 8. In Table 9, the Levene test for equality of variances and the Student’s *t*-test results are obtained.

Figure 7, shows the error graph concerning to the means values analyzed for the spectrum analyzer and the proposed exposimeter, observing that the error graphs are within the same range, and there is no significant differences between both devices. In Figure 8, the correlation graph between the FSH8 spectrum analyzer equipment and the proposed exposimeter at each frequency band of all measurement points from 1 to 7 in Table 6 is shown.

## 4. Discussion

In this study, the regulations established by ICNIRP [3] were not followed, the measurements of the maximum received power levels were considered (not the average). The non-thermal effects are not considered in the establishment of the exposure limits, and the BioInitiative working group, together with other researchers [8,10,12,13,14,15,16], suggest that adverse health effects are observed at low levels of exposure and suggest that using the SAR criterion alone is not the most appropriate for this purpose. Conventional exposimeters are not suitable for differentiating between multiple electromagnetic field sources because their resolution bandwidth is determined by the full desired frequency band to be measured, avoiding the detection of the sources responsible for the greatest contribution of electromagnetic fields [34]. Biological effects due to prolonged radiofrequency radiation exposure are considered to increase when energy peaks are detected.

The proposed system follows the ICNIRP requirement to measure the entire radioelectric spectrum in one second or less [3]. The sweeping time is 901.29 ms, and the switching between each fractal antenna does not cause any delay or uncertainty that could affect the measurements. However, the sweeping time could generate the differences in the power measurements between the spectrum analyzer and the proposed exposimeter, as is observed in Table 6. On the other hand, restricting and performing measurements only in the communication bands used by the base stations (Table 5) will significantly increase the sampling rate, obtaining a better estimation of the received power. However, information from the rest of the radioelectric spectrum could be lost. Conventional exposimeters measure up to 20 frequency bands [33]; therefore, these are very sensitive issues about sweeping time.

Table 7 shows the frequencies that correspond to the maximum power values measured in Table 6. Showing that a high correlation exists at each measurement point. However, there are differences in frequency that are attributed to the different resolution bandwidths of the spectrum analyzer and the proposed exposimeter. The designed system measures all the radioelectric spectrum in multiple narrow bands (300 kHz), allowing the identification of different sources of electromagnetic fields, even within the same communications band, for example, if there are two slots occupied in the same frequency band, the proposed exposimeter will be able to detect both sources, and it will also provide the information of the rest of the spectrum in order to detect the cross-talk effect, which in conventional exposimeters could generate a wrong estimation of the measured power [30,34,35].

Concerning the ICNIRP recommendations, all of the measurements in Table 6 exceed the established limit of 450 µW/cm^2^; however, several of the measurements exceed 0.1 µW/cm^2^, which according to the suggestions of the BioInitiative working group [8], could be related to adverse health effects.

In Table 8 of the Smirnov Kolmogorov test with a 95% confidence level, a significant factor of 0.2 is obtained; therefore, the measurements of both devices have a normal distribution. In Table 9, equal variances in the Levene test and a significant factor of 0.948 for Student’s *t*-test were obtained, indicating there is no significant difference between the measurements between the spectrum analyzer and the exposimeter.

In Figure 8, the Pearson’s correlation factor of 0.9682 is obtained, indicating that there is a strong correlation between the data obtained by the FSH8 spectrum analyzer and the proposed exposimeter, corroborating the results obtained with Student’s *t*-test. Therefore, the proposed exposimeter could be used as a tool to measure electromagnetic fields with reliability comparable to a spectrum analyzer.

## 5. Conclusions

A compact exposimeter system has been proposed, which has similar characteristics to a spectrum analyzer, without losing the advantages of conventional exposimeters, and can be a useful tool for electromagnetic field studies analyzing a far-field. The proposed exposimeter measures the received power in the spectrum between 78 MHz and 6 GHz with a resolution bandwidth of 300 KHz, measuring approximately 19,500 narrow bands; the proposed exposimeter has passed the testing phase compared with spectrum analyzers with a significant factor of 0.948 for Student’s *t*-test and a correlation factor of 0.9682 between the measurements of both devices.

The proposed exposimeter requires one second to measure the entire radioelectric spectrum; therefore, it has less sensitivity to variations in electromagnetic fields. This could be improved by restricting the measurements to the communication bands of interest; however, information from the rest of the radio spectrum is lost. It is important to define the relationship between the communication bands to be measured and the sensitivity to electromagnetic field variations with respect to time.

In future work, it is expected to use several of the proposed devices in electromagnetic field surveys to cover residential areas and provide information as an electromagnetic field sensor network to be used in epidemiological studies in the deployed area.

## Figures and Tables

**Figure 1 sensors-21-07395-f001:**
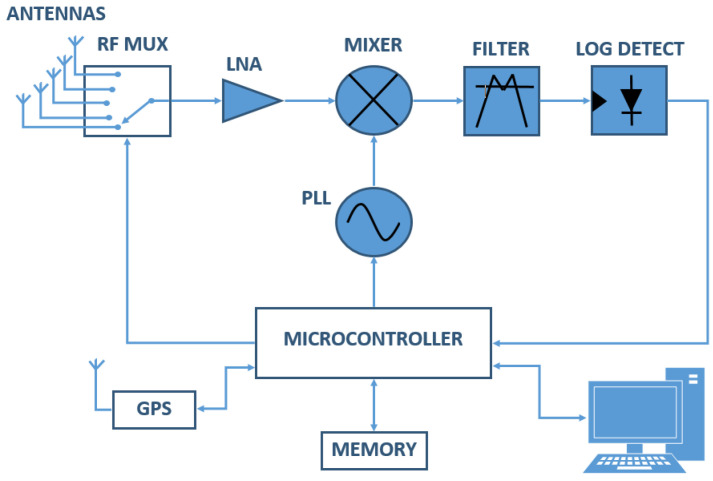
Block diagram of the data acquisition system.

**Figure 2 sensors-21-07395-f002:**
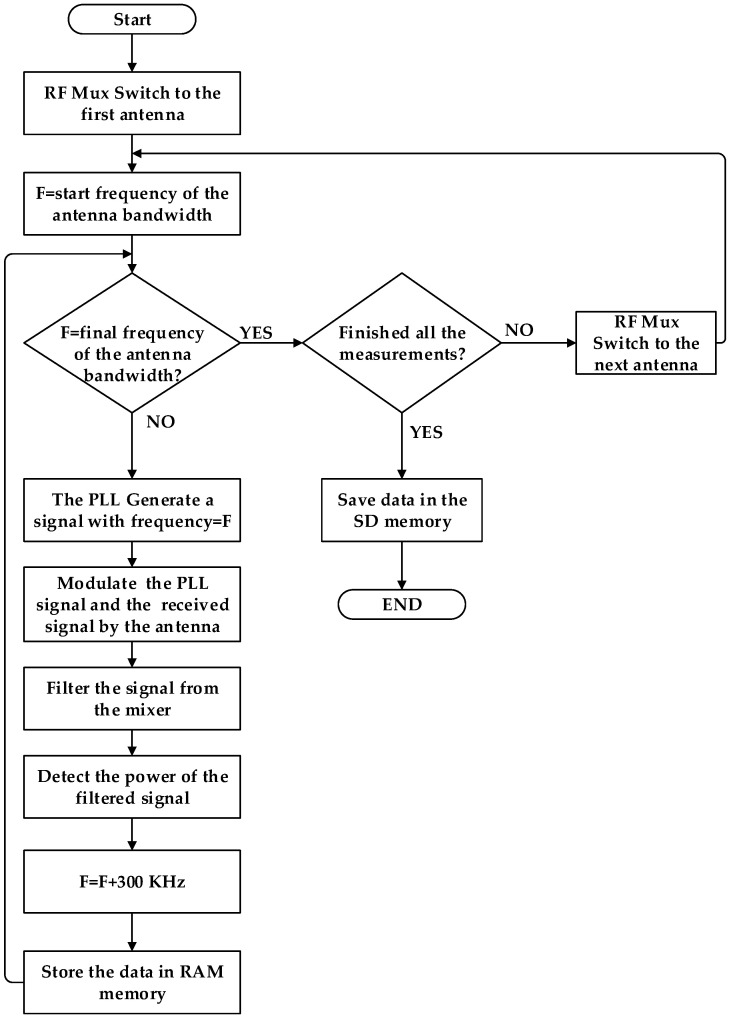
Flow chart for the measurement process.

**Figure 3 sensors-21-07395-f003:**
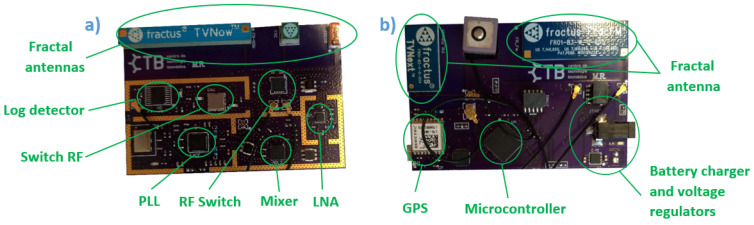
Top view of the proposed system: (**a**) RF PCB; (**b**) control PCB.

**Figure 4 sensors-21-07395-f004:**
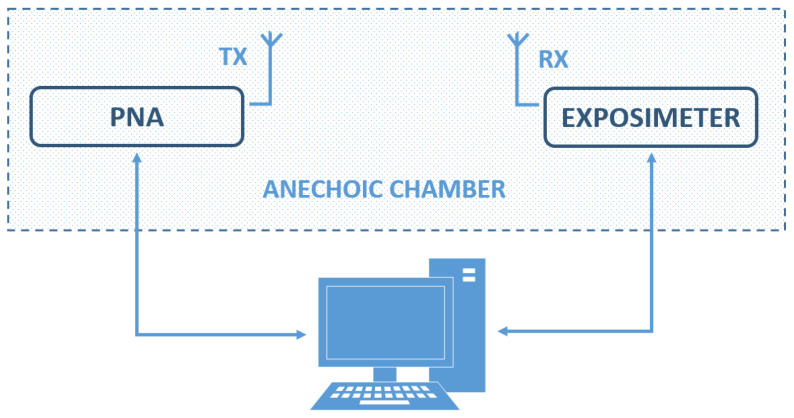
Block diagram of the calibration system.

**Figure 5 sensors-21-07395-f005:**
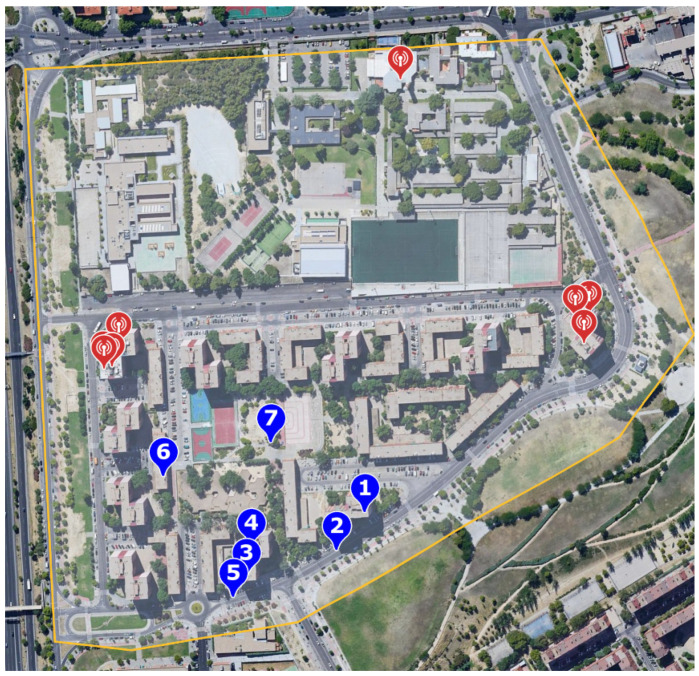
Measurement points in the residential area: In blue, the measurement points, and in red, the location of the base stations.

**Figure 6 sensors-21-07395-f006:**
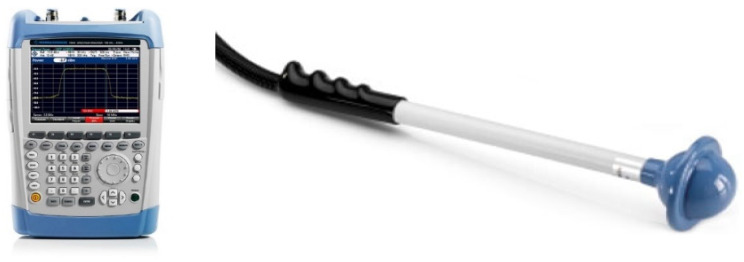
FSH8 portable spectrum analyzer and TSEMF-B2 omnidirectional antenna (Munich, Germany, ROHDE&SCHWARDZ).

**Figure 7 sensors-21-07395-f007:**
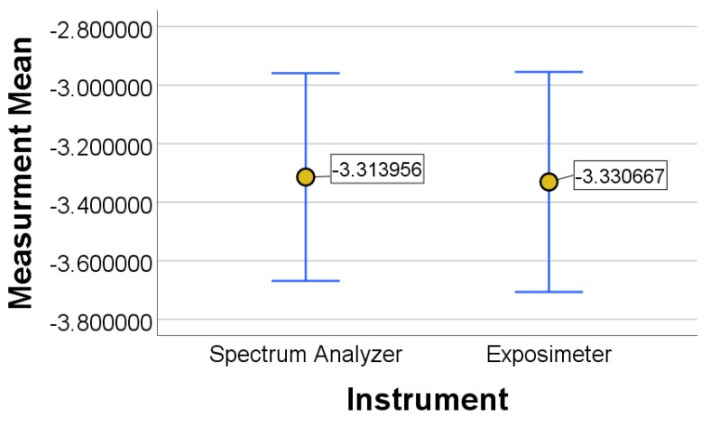
Comparative error bar graph between the FSH8 spectrum analyzer and the proposed exposimeter.

**Figure 8 sensors-21-07395-f008:**
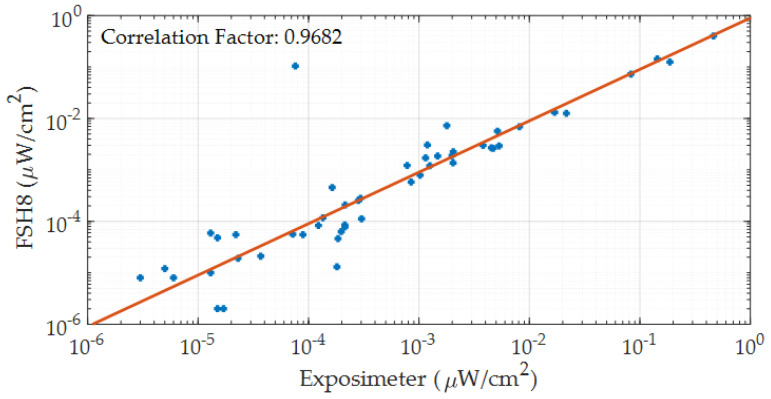
Correlation between the commercial equipment FSH8 and the exposimeter.

**Table 1 sensors-21-07395-t001:** Characteristics of different types of exposimeters.

Frequency Band (MHz)	ESM-140	Expo M-RF	EME Spy-120	EME Spy-121	EME Spy-140	EME Spy-200
FM (87–107)	✗	✓	✓	✓	✓	✓
TV3 (174–223)	✗	✗	✓	✓	✓	✓
TETRA I (380–400)	✗	✗	✓	✓	✓	✓
TETRA II (410–430)	✗	✗	✗	✗	✗	✓
TETRA III (450–470)	✗	✗	✗	✗	✗	✓
TV4&5 (470–770)	✗	✓	✓	✓	✓	✓
LTE 800 DL (791–821)	✗	✓	✗	✗	✗	✓
LTE 800 UL (832–862)	✗	✓	✗	✗	✗	✓
GSM UMTS 900 UL (880–925)	✓	✓	✓	✓	✓	✓
GSM UMTS 900 DL (925–960)	✓	✓	✓	✓	✓	✓
GSM 1800 UL (1710–1785)	✓	✓	✓	✓	✓	✓
GSM 1800 DL (1805–1880)	✓	✓	✓	✓	✓	✓
DECT (1880–1900)	✓	✓	✗	✗	✗	✓
UMTS 2100 UL (1920–1980)	✓	✓	✓	✓	✓	✓
UMTS 2100 DL (2110–2170)	✓	✓	✓	✓	✓	✓
Wifi 2G (2400–2484)	✓	✓	✗	✗	✗	✓
LTE 2600 UL (2500–2570)	✗	✓	✗	✗	✗	✓
LTE 2600 DL (2620–2690)	✗	✓	✗	✗	✗	✓
WiMAX (3300–3900)	✗	✓	✗	✗	✗	✓
Wifi 5G (5150–5850)	✗	✓	✗	✗	✗	✓
Detector	Log	RMS	Log	Log	RMS	RMS
Sampling Interval (s)	0.5–10	3–6000	4–255	4–255	4–255	4–255
Range (V/m)	0.01–70	0.003–5	0.05–10	0.05–10	0.005–10	0.005–5
Lower detection limit (V/m)		0.003	0.05	0.05	0.01	0.005

✗: Does not cover the band ✓: Cover the band.

**Table 2 sensors-21-07395-t002:** Features between exposimeters and spectrum analyzers.

Feature	Spectrum Analyzer	Exposimeters
Full spectrum measurement.	Yes	No (20 bands only)
Detection of multiple sources in the same communications band.	High	No (estimation)
Cross-talk detection	High	No (estimation)
Size	Big	Compact
Cost	High	Low
Portability	Low	High
User-friendliness	Difficult in dynamic environments	Suitable for dynamic environments

**Table 3 sensors-21-07395-t003:** Fractal antennas used and their respective bandwidths.

FRACTUSANTENAS Models	Bandwidth (MHz)
FR01-B3-W-0-055	78–108
FR01-B3-V-0-054	180–220
FR01-B1-S-0-047	470–1675
FR01-S4-250	698–2690
FR05-S1-N0-1-004	2400–5875

**Table 4 sensors-21-07395-t004:** Specifications of the proposed system.

Specification	Value
Operating Range	78 MHz–6 GHz
Sampling Rate	20k samples/s
Dynamic Range	90 dB
Minimum Input Power	−70 dBm
Maximum Input Power	20 dBm
Sweeping time	902 ms
Time memory space	1 week
Current Consumption	450 mA
Battery autonomy	hours

**Table 5 sensors-21-07395-t005:** Frequency bands of the telephone base stations near the measurement points.

Frequency Band	Frequency (MHz)
GSM 800	791–862
GSM 900	880–960
GSM 1800	1710–1879
GSM 1900	1900–1980
GSM 2100	2110–2170

**Table 6 sensors-21-07395-t006:** Highest power peaks at each measurement point and correlation factor.

Frequency Band		MP ^1^ 1 (µW/cm^2^)	MP ^1^ 2 (µW/cm^2^)	MP ^1^ 3 (µW/cm^2^)	MP ^1^ 4 (µW/cm^2^)	MP ^1^ 5 (µW/cm^2^)	MP ^1^ 6 (µW/cm^2^)	MP ^1^ 7 (µW/cm^2^)
GSM 800	SA ^2^	0.005136	0.005321	0.000215	0.001253	0.000037 *	0.002038	0.1437
	E ^3^	0.005661	0.002927	0.000208	0.001201	0.000021 *	0.001364	0.1431
GSM 900	SA ^2^	0.000017	0.000015	0.000135	0.000781	0.001148	0.003805	0.08302
	E ^3^	0.000002 *	0.000002 *	0.000117	0.001221	0.001709	0.002987	0.0729
GSM 1800	SA ^2^	0.000006 *	0.02161	0.004523	0.000013 *	0.000003 *	0.001477	0.4644
	E ^3^	0.000008 *	0.01253	0.00268	0.00001 *	0.000008 *	0.001852	0.4025
GSM 1900	SA ^2^	0.01691	0.004712	0.000022 *	0.000283	0.000123	0.000013 *	0.000015 *
	E ^3^	0.01309	0.00263	0.000055 *	0.000253	0.000083	0.000059 *	0.000048 *
GSM 2100	SA ^2^	0.000181	0.000023 *	0.000198	0.000214	0.000005 *	0.000213	0,187
	E ^3^	0.000013 *	0.000019 *	0.000063 *	0.000078 *	0.000012 *	0.000085 *	0.1252
Wifi 2G	SA ^2^	0.001993	0.001788	0.000164	0.000302	0.000296	0.001192	0.000076 *
	E ^3^	0.001891	0.007244	0.000455	0.000112	0.000281	0.003039	0.1043
Wifi 5G	SA ^2^	0.000072 *	0.002044	0.000185	0.00085 *	0.001024	0.008116	0.000089 *
	E ^3^	0.000056 *	0.002246	0.000046 *	0.00058 *	0.000786	0.006891	0.000055 *
Correlation factor		0.9916	0.9195	0.9262	0.9290	0.8971	0.8970	0.9680

* Noise. ^1^ MP: Measuring Point. ^2^ SA: Spectrum Analyzer. ^3^ E: proposed Exposimeter.

**Table 7 sensors-21-07395-t007:** Frequencies of the power peaks of each measurement point and its correlation factor.

Frequency Band		MP ^1^ 1 (MHz)	MP ^1^ 2 (MHz)	MP ^1^ 3 (MHz)	MP ^1^ 4 (MHz)	MP ^1^ 5 (MHz)	MP ^1^ 6 (MHz)	MP ^1^ 7 (MHz)
GSM 800	SA ^2^	784	776	809	809	824 *	809	809
	E ^3^	745	736	810	800	738 *	800	809
GSM 900	SA ^2^	935 *	927 *	927	852	893	935	927
	E ^3^	949 *	957 *	928	851	864	951	928
GSM 1800	SA ^2^	1841 *	1718	1768	1777 *	1815 *	1836	1853
	E ^3^	1865 *	1727	1789	1725 *	1731 *	1847	1851
GSM 1900	SA ^2^	1886	1895	1931 *	1861	1903	1903 *	1908 *
	E ^3^	1882	1888	1975 *	1892	1851	1947 *	1969 *
GSM 2100	SA ^2^	2135 *	2144 *	2122	2122	2158 *	2130	2147
	E ^3^	2169 *	2169 *	2138 *	2101	2169 *	2141	2141
Wifi 2G	SA ^2^	2416	2441	2416	2438	2475	2467	2441
	E ^3^	2469	2448	2461	2441	2461	2436	2481
Wifi 5G	SA ^2^	5735 *	5226	5192	5293 *	5226	5277	5345 *
	E ^3^	5775 *	5215	5623 *	5585 *	5218	5270	5658 *
Correlation factor		0.9999	0.9999	0.9984	0.9986	0.9986	0.9987	0.9987

* Noise. ^1^ MP: Measuring Point. ^2^ SA: Spectrum Analyzer. ^3^ E: proposed Exposimeter.

**Table 8 sensors-21-07395-t008:** Kolmogorov Smirnov normality test.

Instrument	N	Factor Significance
Spectrum analyzer	49	0.200
Exposimeter	49	0.200

**Table 9 sensors-21-07395-t009:** Levene test for equality of variances and Student’s *t*-test.

		Equal Variances
Levene Test	Significance Factor	0.525
Student’s *t*-test	Significance Factor	0.948
	Mean Difference	0.01671
	Standard error difference	0.2569
	95% CI ^1^ Lower	−0.4933
	95% CI ^1^ Higher	0.5267

^1^ 95% CI: 95% Confidence Interval.

## Data Availability

The data presented in this study are available on request from the corresponding author.

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
