# Peer review of "Compact Exposimeter Device for the Characterization and Recording of Electromagnetic Fields from 78 MHz to 6 GHz with Several Narrow Bands (300 kHz)"

_sensors, 2021, doi:10.3390/s21217395_

Round 1

Reviewer 1 Report

My comments are in the attached file.

Reviewer 2 Report

The paper presents a novel compact exposimeter device for recording EMF up to 6GHz, including the design and validation of the exposimeter.

Why do authors mean that exposimeters cannot detect multiple sources in the same band? Does it depend on the resolution bandwidth of device?

In the Block diagram of the data acquisition system, why there are four antennas? Is there more detailed description on the fractal antennas used, like the radiation pattern of each single antenna, since it is supposed to mimic the omnidirectional antenna.

How does the RF mux switch between different ports? Does it switch to different ports in each band, or it finishes all bands then switch to the next band? Please add flow charts of steps in each single measurement, and comment on the switching time consumed in compare with total time of 1s to cover all bands.

Please comment on the applicability of such developed exposimeter in driving test. Will the switching between each port of fractal antenna cause delay and uncertainty in the measurement?

What is the noise floor for each band? Since authors use same RBW for both high frequency and low frequency. And what is the sweeping time?

In the calibration, why the authors chose 2 meters to be the distance for all frequencies (78 MHz to 6GHz)? Since for 78MHz, 2 meters already lies in near-field if we adopt wavelength as near-field distance.

With the antenna directly attached to the PCB, have the authors considered shadowing caused by the PCB and the expert personnel in the measurement? Hand-hold or put it on a platform, what is the height to the ground? Is there mobile phone is use (e.g., file uploading or stream video) to stimulate the signal from BS?

Why authors chose maximum received power instead of average power (or distribution of power) in the validation of proposed design? As maximum power may not be very representative.

In Table 6, what is the stage? repeated measurement at different time period? Then why there are some missing of measurements in the table? Please comment on the noise level for both designed exposimeter and FSH8 in each band, since if there is no signal from certain frequency band, it could be just nosie.

What is PDP-EMF in Figure 6? Figure 7 needs logarithm plot in order to see more details at low values.

Please correct all the symbols with superscript or subscript in the paper, e.g., line 164, 165, 171, 177, 181, etc.

Line 164-165, too much ‘and’ in the sentence, please rewrite it. “Figure 2B” to “Figure 2b” and please change the order of first PCB and second PCB, to make it consistent with a) and b) in the Figure.

Line 189-191, too much ‘and’ in the sentence, please rewrite it.

Line 94: “chipper” to “cheaper”

Line 184, “communicate” to “communicating”

Line 200-201, the dynamic power measurement range is 100dB or 90dB ?

Line 282, “analyzer and TSEMF-B2” to “analyzer with TSEMF-B2”

Round 2

Reviewer 2 Report

The paper can be accepted in the present form.